# De novo discovery of traits co-occurring with chronic obstructive pulmonary disease

Evgeniia Golovina[1], Tayaza Fadason[1,2], Rachel K Jaros[1], Haribalan Kumar[3], Joyce John[3], Kelly Burrowes[3], Merryn Tawhai[3], Justin M O'Sullivan[1,2,4,5,6]

**Chronic obstructive pulmonary disease (COPD) is a heterogeneous group of chronic lung conditions. Genome-wide association studies have identified single-nucleotide polymorphisms (SNPs) associated with COPD and the co-occurring conditions, suggesting common biological mechanisms underlying COPD and these co-occurring conditions. To identify them, we have integrated information across different biological levels (i.e., genetic variants, lung-specific 3D genome structure, gene expression and protein–protein interactions) to build lung-specific gene regulatory and protein–protein interaction networks. We have queried these networks using disease-associated SNPs for COPD, unipolar depression and coronary artery disease. COPD-associated SNPs can control genes involved in the regulation of lung or pulmonary function, asthma, brain region volumes, cortical surface area, depressed affect, neuroticism, Parkinson's disease, white matter microstructure and smoking behaviour. We describe the regulatory connections, genes and biochemical pathways that underlay these co-occurring trait-SNP-gene associations. Collectively, our findings provide new avenues for the investigation of the underlying biology and diverse clinical presentations of COPD. In so doing, we identify a collection of genetic variants and genes that may aid COPD patient stratification and treatment.**

## Introduction

Chronic obstructive pulmonary disease (COPD) is a heterogeneous group of chronic lung conditions that are characterized by persistent respiratory symptoms and airflow limitation due to airway and/or alveolar abnormalities (2020 Global Initiative for Chronic Obstructive Obstructive Lung Disease, 2021). These abnormalities are caused by a combination of distinct pathophysiological processes that result in diverse clinical presentations, responses to treatment, and patterns of progression. According to the World Health Organization, COPD accounted for more than 3.23 million deaths in 2019 and remains the third leading cause of death worldwide (2020 Global Initiative for Chronic Obstructive Obstructive Lung Disease, 2021).

Given the widespread exposure to the environmental factors (e.g., smoking, indoor and outdoor air pollution, childhood respiratory infections) that contribute to the development of COPD, it is striking that most individuals will never develop COPD. The variance in individual susceptibility to COPD can be partly explained by genetic factors. The estimated genetic heritability of COPD ranges from 20% to 40% (Zhou et al, 2013; Gim et al, 2020; Stolz, 2020).

Co-occurring conditions are widely recognised as impacting on COPD patient outcomes (Cavaillès et al, 2013; Burke & Wilkinson, 2021). As such, a better understanding of COPD co-occurring conditions is essential to enable effective management, therapeutic optimization and reduce the costs of managing COPD patients (Mannino et al, 2015). Epidemiological and genetic studies have reported that beyond respiratory impairment COPD-associated co-occurring conditions include coronary artery disease (CAD), lung cancer, osteoporosis, mental health problems such as anxiety, unipolar depression (UD), Alzheimer's disease (AD), and Parkinson's disease (PD) (Cavaillès et al, 2013; Li et al, 2015; Ställberg et al, 2018; Xia et al, 2020; Burke & Wilkinson, 2021; Carmona-Pírez et al, 2021; Martucci et al, 2021). The presence of these conditions in COPD patients indicates that common or interacting biological mechanisms underlie these conditions.

To date, genome-wide association studies (GWASs) have identified common single nucleotide polymorphisms (SNPs) that are associated with COPD, or its individual co-occurring conditions (Sakornsakolpat et al, 2019; Shrine et al, 2019; Zhu et al, 2019; Kim et al, 2021b). Most of the COPD-associated SNPs are located within the non-coding genome. Therefore, the impacts that these SNPs have on the biological pathways and processes underlying the development of COPD remain unclear. It is possible that the COPD-associated SNPs mark regulatory regions (i.e., expression quantitative trait loci [eQTLs]) that are associated with tissue-specific gene expression. eQTLs can interact with their target genes in three dimensions, forming spatial eQTL–gene regulatory connections that span the genome (e.g., cis, ≤1 Mb on the same chromosome;

[1]Liggins Institute, University of Auckland, Auckland, New Zealand   [2]Maurice Wilkins Centre, University of Auckland, Auckland, New Zealand   [3]Auckland Bioengineering Institute, University of Auckland, Auckland, New Zealand   [4]MRC Lifecourse Epidemiology Unit, University of Southampton, Southampton, UK   [5]Garvan Institute of Medical Research, Sydney, Australia   [6]Singapore Institute for Clinical Sciences, Agency for Science, Technology and Research (A*STAR), Singapore, Singapore

Correspondence: justin.osullivan@auckland.ac.nz

trans-intrachromosomal, >1 Mb on the same chromosome; or trans-interchromosomal, between different chromosomes). These spatial interactions are cell and tissue type–specific (GTEx Consortium, 2020; Halow et al, 2021). As lung is the primary affected tissue in COPD, integrating lung-specific spatial chromatin interactions and eQTL information may help us understand how SNPs impact biological pathways that increase an individual's risk of developing COPD.

Little is known about functional relationships between genes and phenotypes in the lung. However, gene regulation is widely understood to occur through the combinatorial action of regulatory elements, transcription factors and genes within complex networks (i.e., gene regulatory network [GRN]) (Buenrostro et al, 2018; Chen et al, 2021a, 2021b; Zaied et al, 2022 Preprint). Moreover, genes encode proteins that physically interact with each other to form a complex protein–protein interaction network (PPIN) that responds to biological and environmental signals. Here, we integrated COPD-associated SNPs with: (1) information on the genome organization within the lung; and (2) lung-specific eQTL information to identify genes that are spatially regulated within the lung tissue. We integrated information across a lung-specific GRN and PPIN to identify conditions that were co-occurring with COPD. Collectively, our results highlight potential regulatory mechanisms and pathways important for COPD etiology. These results open a new avenue towards understanding the diverse clinical presentations of COPD and patient stratification.

# Results

## COPD-associated SNPs mark putative regulatory regions in the lung

COPD-associated SNPs ($P < 5 \times 10^{-8}$, n = 263) were downloaded from the GWAS Catalog (Tables S1 and S2) and run through the CoDeS3D pipeline (Fig 1A). ~96% of the identified eQTLs were located within non-coding genomic regions, with 66.02% and 18.45% of them being intronic and intergenic, respectively (Fig S1A and B [wANNOVAR annotation], Table S2 [original GWAS Catalog annotation]). Analysis of these SNPs using the CoDeS3D (Fadason et al, 2018) pipeline (Fig 1A) identified 103 eQTLs and 107 genes that are involved in 151 significant (FDR < 0.01) eQTL–gene interactions within the lung (Fig S1 and Table S3). Most COPD-associated eQTLs (n = 67) are involved in one-to-one, 26 eQTLs—in one-to-two and 8 eQTLs—in one-to-three eQTL–gene regulatory interactions (Fig S1C and Table S3). Only two eQTLs (i.e., rs2277027 and rs9435731) were associated with the regulation of ≥4 genes (i.e., ADAM19, CTB-109A12.1, CTB-47B11.3, CYFIP2 and ATP13A2, CROCC, MFAP2, RP1-37C10.3, respectively; Fig S1C and Table S3). Most of the identified eQTL–gene regulatory interactions (n = 148) were cis-acting (Fig S1D and Table S3). One trans-intrachromosomal (i.e., rs2077224-NAV2) and two trans-interchromosomal (i.e., rs12894780-LIPC and rs2128739-KALRN) eQTL–gene interactions were identified within the lung (Fig S1D and Table S3). Collectively, COPD-associated eQTLs are associated with changes in transcription levels of 84 protein-coding genes, 22 non-coding RNA genes and one pseudogene (Fig S1E).

## COPD-associated genes are enriched for diverse biological processes in the lung

Functional gene ontology (GO) enrichment analysis identified metabolic, behavioural, regulatory and protein modification processes (e.g., "phosphorus metabolic process," "behavioral response to nicotine," "regulation of postsynaptic membrane potential," "protein acetylation," and "protein acylation") as being significantly enriched (FDR < 0.05) enriched within the 107 COPD-associated genes (Table S4 and Fig S4). These 107 genes encoded proteins that formed nine COPD-associated lung-specific protein–protein interaction subnetworks (Fig 2). Pathway analysis of these 107 COPD-associated genes identified biological pathways that were enriched (FDR < 0.05) for regulation of actin cytoskeleton, insulin signaling and resistance, focal adhesion, phagosome, immune processes, infections and diseases, alcoholism, long-term depression (Table S5).

## COPD has associations with co-occuring traits in the lung, brain, and blood

Patients with COPD often also suffer from cardiovascular disease, osteoporosis, lung cancer, sleep disorders and mental health problems (Burke & Wilkinson, 2021; Carmona-Pírez et al, 2021). Yet the biology of these interactions is rarely known. The multimorbid3D algorithm was used to integrate COPD-associated genes, the lung-specific PPIN, the lung-specific GRN and all catalogued GWAS SNP-trait associations (30/03/2022) to identify co-occurring conditions and potential regulatory mechanisms underlying these associations with COPD (Fig 1B). We identified 39 GWAS traits that are significantly (FDR ≤ 0.05) enriched for eQTLs that target the COPD-eQTL associated genes ("level 0"; Fig 3 and Tables S6 and S7). Most of the level 0 co-occurring traits were "lung-" (i.e., COPD, lung function, pulmonary function, post bronchodilator FEV1, asthma), or "mood/brain-related" (i.e., brain region volumes, cortical surface area, depressed affect, neuroticism, PD, white matter microstructure, smoking behaviour). eQTLs that regulate genes encoding proteins within levels 1–4 of the expanded COPD lung protein interaction network were enriched within traits that have and have not been previously recognized as being co-occurring with COPD (Fig 3).

## COPD shows associations with lung function, CAD, AD, and brain region volumes

Proteins encoded by four COPD-associated genes (i.e., MSL1, MOCS2, NUPR1, and SGF29) form a PPI subnetwork (level 0). These genes are associated with eQTLs linked to COPD and lung function (Fig 4A). Notably, MSL1, MOCS2, NUPR1, and SGF29 interact with proteins encoded by three genes that are associated with post bronchodilator FEV1, atopic asthma and AD (Figs 4A and S5A).

COPD-associated SSH2 and TESK2 genes form another PPI subnetwork (level 0). Within this subnetwork they are associated with eQTLs linked to COPD, lung functioning, brain region volumes and CAD (Fig 4B). SSH2 interacts with SSH1, SSH3, LIMK1, CFL1, CFL2, PPP3CA, and PPP3CC proteins at the "level 1" (Fig S5B), but there were no GWAS traits identified for the eQTLs that are associated with expression changes of the genes encoding these proteins.

Reversing the analysis, using CAD-associated (n = 804) and UD-associated (n = 932) SNPs ($P < 5 \times 10^{-8}$) confirmed COPD was

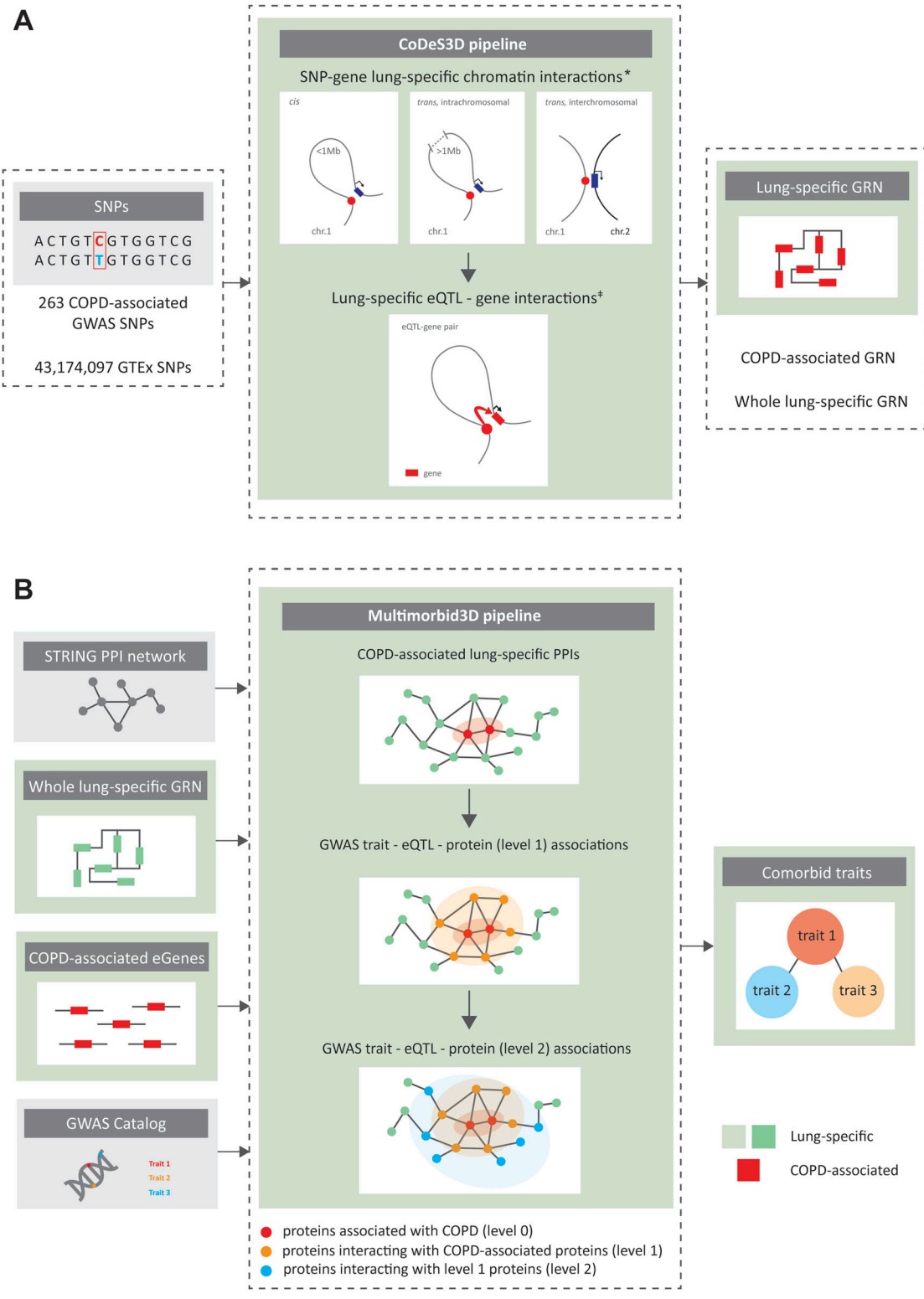

**Figure 1. Overview of the analytical pipelines used in this study.**
**(A)** The CoDeS3D pipeline was used to identify lung-specific gene regulatory networks (GRNs). First, 263 GWAS SNPs associated ($P < 5 \times 10^{-8}$) with COPD were run through the CoDeS3D pipeline to identify 151 spatial eQTL–gene regulatory interactions in the lung (COPD-associated lung-specific GRN). Next, all GTEx SNPs (MAF ≥ 0.05, n = 43,174,097) were downloaded from dbGaP (Table S1) and analysed using CoDeS3D to identify "all" significant lung-specific spatial eQTL–gene regulatory interactions (whole lung-specific GRN). The resultant whole lung-specific GRN is comprised of 873,133 spatially constrained regulatory interactions involving 740,028 eQTLs and 15,855 genes (Figs S2 and S3). **(B)** The Multimorbid3D pipeline was used to identify potential co-occurring conditions associated with COPD. * Hi-C datasets for primary lung cells were obtained from Schmitt et al (2016) (GEO accessions: GSM2322544 and GSM2322545). ‡ eQTL datasets for lung was obtained from GTEx v8 (dbGaP accession: phs000424.v8.p2).

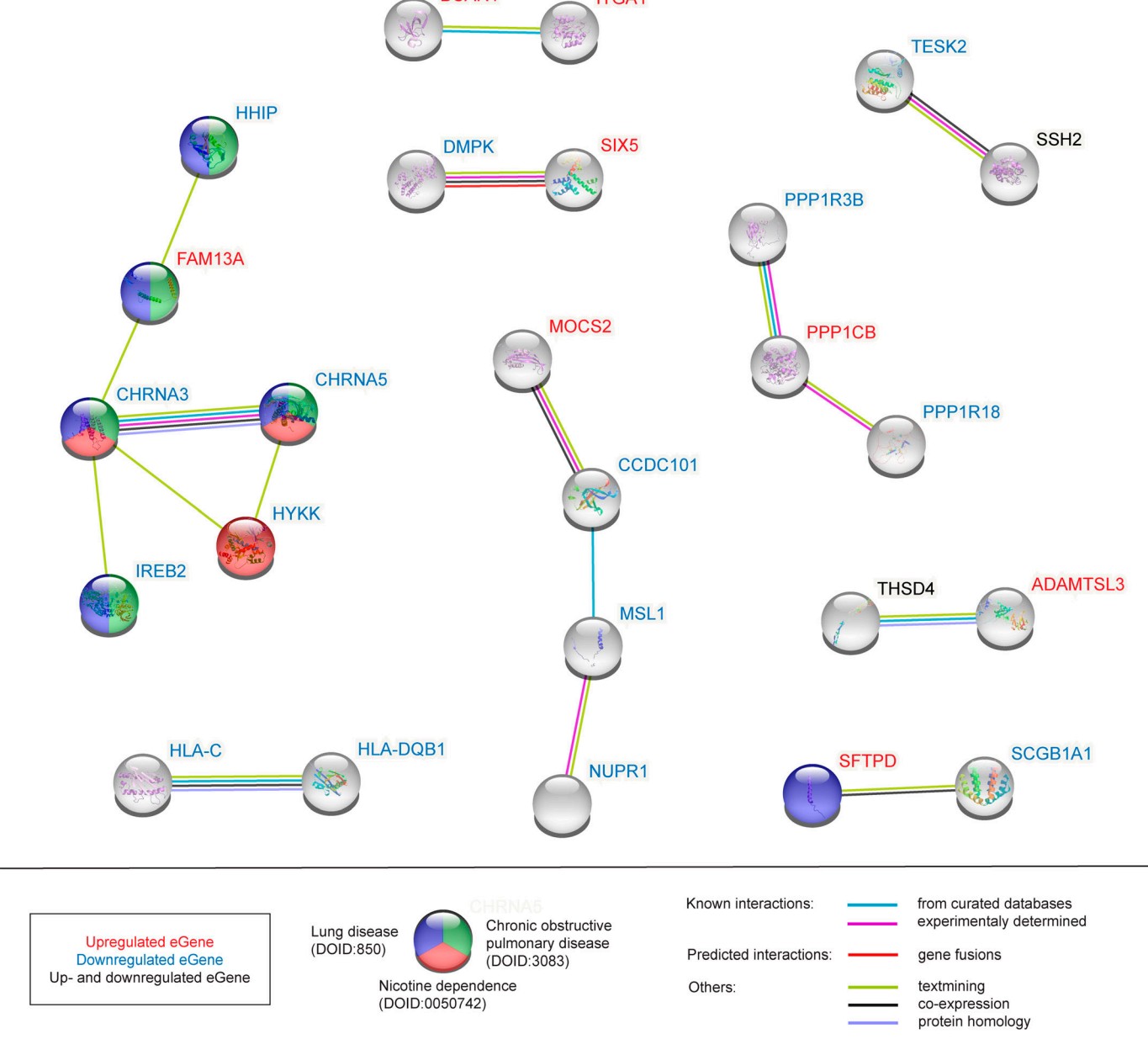

**Figure 2. COPD-associated protein–protein interactions.**
Green colour, the protein encoding gene is associated with COPD (DOID:3083); blue colour, the gene is associated with lung disease (DOID:850); red colour, the gene is associated with nicotine dependence (DOID:0050742) according to the DISEASES database (https://diseases.jensenlab.org/, 06/10/2022). Red text, the eQTLs is associated with up-regulation of the gene; blue text, the eQTL is associated with down-regulation of the gene transcript; black text, eQTLs are associated with up- and down-regulation of the gene transcript levels.

co-occurring with CAD ("level 0," Fig S6 and Table S8). By contrast, analysis of UD identified general lung function (FEV1/FVC), asthma and "mood/brain-related" (i.e., bipolar disorder, depression, autism spectrum disorder, and schizophrenia) as being linked to UD-associated genes (Fig S7 and Table S8).

# Discussion

We integrated genes that are targeted by spatially constrained COPD-associated eQTLs with a lung-specific GRN, lung-specific

PPIN and all GWAS SNP-trait associations to identify traits that are co-occurring with COPD. The results of this integration provide insights into the regulatory mechanisms underlying these associations. We identified co-occurring traits that have been previously linked to COPD (e.g., lung function, asthma, depressed affect, CAD, AD, smoking behaviour, PD [Cavaillès et al, 2013; Li et al, 2015; Ställberg et al, 2018; Xia et al, 2020; Burke & Wilkinson, 2021; Carmona-Pírez et al, 2021; Martucci et al, 2021]) and those that have not (brain region volumes and white matter microstructure). We contend that the eQTLs we identified, as impacting on COPD and its co-occurring traits, represent the population-based genetic burden

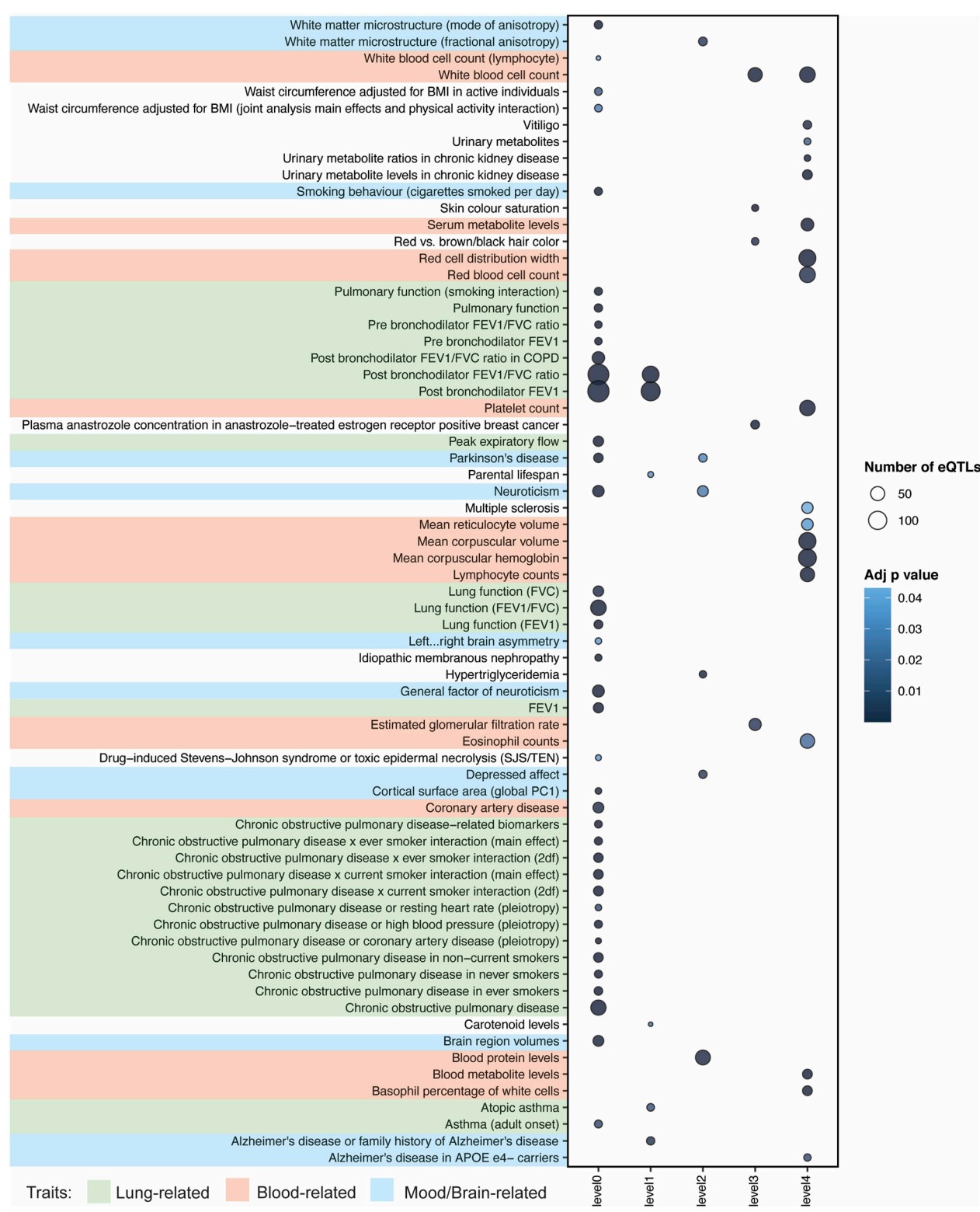

**Figure 3. Network analysis identified co-occurring conditions that are associated with COPD.**
We identified 39 GWAS traits that are enriched (FDR ≤ 0.05) for eQTLs associated with COPD-eQTL target genes (level 0). Most of these co-occurring traits are "lung-related" (i.e., COPD, lung function, pulmonary function, post bronchodilator FEV1, and asthma) and "mood/brain-related" (i.e., brain region volumes, cortical surface area, depressed affect, neuroticism, Parkinson's disease, white matter microstructure, and smoking behaviour). Genes interacting with COPD-associated genes (level 1) within LSPPIN are regulated by eQTLs that have previously been associated with Alzheimer's disease, atopic asthma, and post bronchodilator FEV1 or FEV1/FVC ratio. Level 2-genes are only regulated by eQTLs previously associated with "mood/brain-related" (e.g., depressed affect, neuroticism, and Parkinson's disease) or "blood-related" (e.g., blood protein levels) traits. Genes within levels 3 and 4 are mostly associated with eQTLs enriched within "blood-related" traits.

## A "NUPR1-MSL1-SGF29-MOCS2" subnetwork

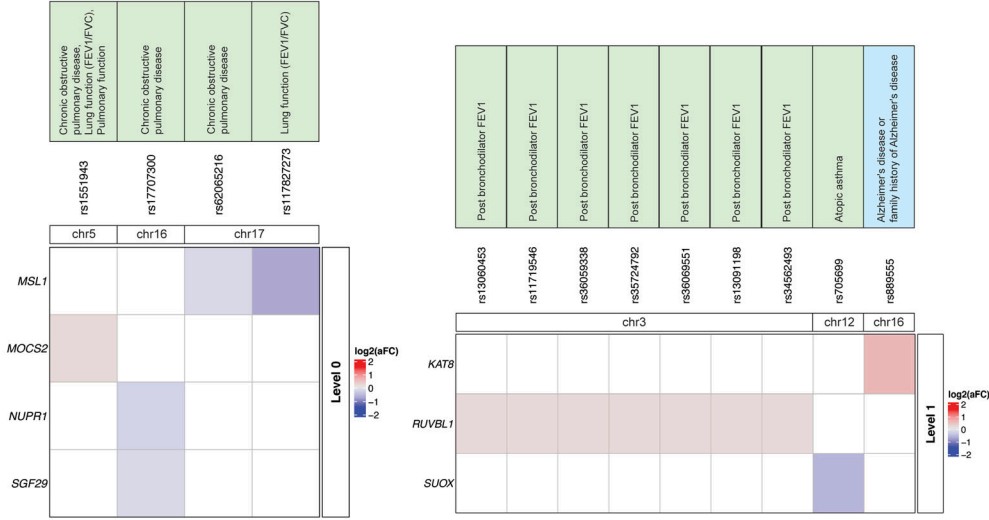

**Figure 4. Identified trait-eQTL–gene associations for two COPD-associated PPI subnetworks (NUPR1-MSL1-SGF29-MOCS2 and TESK2-SSH2).**
**(A)** Within the "NUPR1-MSL1-SGF29-MOCS2" PPI subnetwork four genes (i.e., *MSL1*, *MOCS2*, *NUPR1*, and *SGF29*) are associated with eQTLs linked to COPD and lung functioning. Proteins encoded by these four genes interact with the products of three genes ("level 1") that are associated with post bronchodilator FEV1, atopic asthma, and Alzheimer's disease. **(B)** *SSH2* and *TESK2* within the "TESK2-SSH2" PPI subnetwork are associated with eQTLs linked to COPD, lung functioning, brain region volumes, and CAD. There were no trait-eQTL–gene associations identified within level 1.

## B "TESK2-SSH2" subnetwork

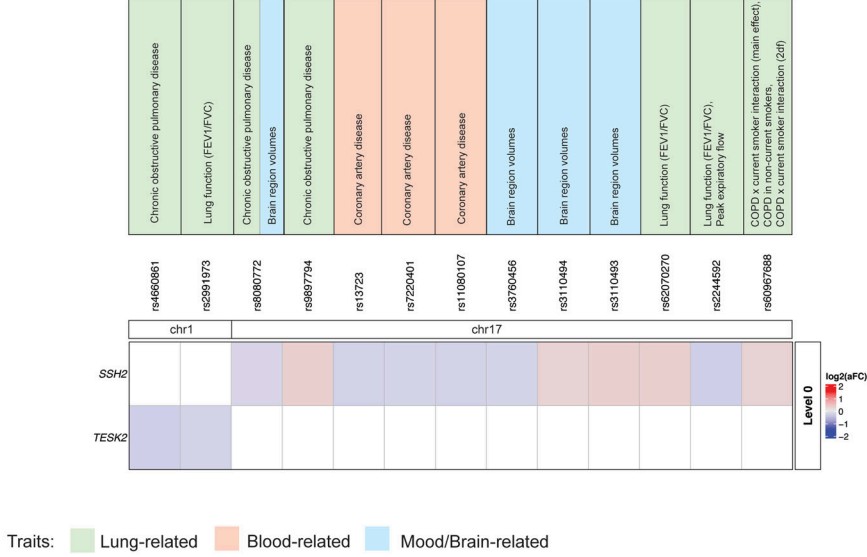

that may contribute to an individual's risk of developing COPD and its diverse clinical presentations.

The results of this study should be interpreted in view of its strengths and limitations. The main strength of this study is the integration of independent datasets: lung-specific 3D genome structure (Schmitt et al, 2016), common GWAS SNPs, genotypes and gene expression data (Aguet et al, 2019 *Preprint*), and protein-protein interactions (PPIs) (Szklarczyk et al, 2019). Integrating these datasets enabled the identification of the impact of spatially constrained COPD-associated eQTLs on genes and biological pathways that link to co-occurring conditions. Indeed it is possible that the "total" genetic burden we identified can be used to stratify

COPD and its multiple clinical presentations, which have previously led to questions about the validity of classifying it as a single diagnostic category (Sakornsakolpat et al, 2019; Corlateanu et al, 2020; Alabi et al, 2021). However, this study also has several limitations. Firstly, this study was focused on the regulatory roles of common genetic variants (MAF ≥ 0.05) ignoring the impacts of other genetic (e.g., rare SNPs) and environmental factors, which will undoubtedly contribute to the risk of COPD and its co-occurring conditions. Secondly, we focused on the extended PPIN within the lung GRN, as the lung represents the primary affected tissue in COPD. However, it is possible that genetic variation will impact on COPD risk through other tissues (e.g., blood) (Burke & Wilkinson, 2021). Thirdly, the

tools and datasets used in this study are potentially biased (e.g., lung-specific eQTL and Hi-C data were not obtained from identical samples). Furthermore, the identification of the co-occurring conditions was limited to the traits that have been studied and were present in the GWAS Catalog. Finally, mapping to Ensembl gene identifiers potentially causes a loss of data specificity, since alternative splicing typically produces multiple transcripts and protein variants.

Co-occurring conditions are commonly associated with COPD and increase the risk of hospitalisation (Schnell et al, 2012). Our analysis of co-occurring conditions has identified risk variants and protein interactions that connect COPD with smoking (Hopkinson, 2022), asthma (Maselli & Hanania, 2018), CAD (Xia et al, 2020), lung cancer (Parris et al, 2019), multiple sclerosis (Egesten et al, 2008), kidney failure (Trudzinski et al, 2019), UD (Li et al, 2019a), AD (Wang et al, 2019), PD (Li et al, 2015), and personality traits (e.g., neuroticism) (Chetty et al, 2017; Terracciano et al, 2017; Caille et al, 2021). These conditions have been previously reported as being co-occurring conditions in COPD patients but the biological basis of the connections was unknown.

Brain region volumes, white matter microstructure, left-right brain asymmetry and cortical surface area (global PC1) were additional traits that we have identified as being co-occurring with COPD. Notably, patterns of brain structural alteration have been reported in COPD with different levels of pulmonary function impairment and cognitive deficits (Yin et al, 2019; Wang et al, 2020). This suggests that COPD patients may exhibit progressive structural impairments in both the grey and white matter, along with impaired levels of lung function (Yin et al, 2019; Wang et al, 2020). In addition to the trait associations, the results of our study provide putative evidence for the existence of genetic and biological connections between these traits (Higbee et al, 2021). In so doing our results stop short of providing causal evidence for these putative connections and two sample or multivariable Mendelian randomization (e.g., Transcriptome-Wide Mendelian Randomization) or mediation analyses are required to clarify which if any of the interactions are causal (Russ et al, 2021; Porcu et al, 2019).

Impaired lung function as measured by forced vital capacity or forced expiratory volume in the first second (FEV1) has been previously associated with insulin resistance (Sagun et al, 2015; Piazzolla et al, 2017; Machado et al, 2018; Kim et al, 2021a). The evidence that supports the existence of a relationship between COPD and insulin resistance is complex. Our independent identification that *PPP1CB* and *PPP1R3B* are COPD-associated genes is notable because these genes are directly involved in controlling glycogen synthesis and glucose homeostasis in insulin signaling (Li et al, 2019b) and have been linked to glycaemic traits (Niazi et al, 2019).

We also identified several COPD-associated genes enriched for immune processes and immune-related diseases (e.g., *HLA-C* and *HLA-DQB1*—antigen processing and presentation, *HLA-DQB1*—type I diabetes mellitus, influenza A, autoimmune thyroid disease) (Table S5). This finding is consistent with observations that proteins of the major histocompatibility complex of classes I and II (HLA-I and HLA-II) have been identified as potential markers of progression of systemic and local inflammation in patients with COPD (Kubysheva et al, 2018). Specifically, an increase in the level of HLA-I and HLA-II

molecules in the exhaled breath condensate as well as an elevated serum level of HLA-II is observed in COPD patients when compared with healthy volunteers (Kubysheva et al, 2018). Previous studies have also identified elevated levels of plasma TGF-*β*, an important regulator of lung and immune system development, in COPD patients compared with healthy controls (Mak et al, 2009; Verhamme et al, 2015). Despite the apparent correlative support between our results and previously published fundings, the evidence for a role for the insulin resistance pathway and immune system in COPD remains putative until proven empirically.

COPD-associated eQTLs target cholinergic pathway genes (e.g., *CHRM1* and *CHRM3*) that have previously been implicated as important susceptibility loci for lung diseases (e.g., asthma and COPD) (Palmberg et al, 2018; Rajasekaran et al, 2019). The cholinergic pathway mediated by the parasympathetic neurotransmitter, acetylcholine, is a predominant neurogenic mechanism contributing to bronchoconstriction (Ward, 2022). Notably, changes in the parasympathetic neuronal control of airway smooth muscle have been shown to increase bronchoconstriction in response to vagal stimulation, leading to airway hyperresponsiveness (Ward, 2022). At the biological level, these findings emphasize the effects that COPD-associated SNPs may have on the regulation of genes within specific biological pathways (e.g., through creating an imbalance in the concentration of specific proteins), which, in turn, can be associated with an increased risk of disease.

Integrative network-based methods have been used to explore complex SNP-gene interactions that can impact functional biological pathways and lead to complex phenotypes (Platig et al, 2016; Chen et al, 2021b; Zhu et al, 2021). The CONDOR algorithm was applied to study eQTLs in COPD (Platig et al, 2016). In brief, CONDOR built a bipartite network linking eQTLs and genes in 52 communities. 30 SNPs associated with COPD (Cho et al, 2014) mapped to three of these communities. Notably, despite only one gene (*KANSL1*) being identified in Platig et al (2016) and our current study, the loci involved were enriched in conserved biological processes including Alzheimer's, Parkinson's, and Asthma and immune responses. This is despite fundamental difference in the methods (e.g., eQTLs vs spatially constrained eQTLs, and the inclusion of the expanded protein network as additional information in multimorbid3D) and updates in the GWAS catalog that occurred between 2016 and 2022. As such, the apparent convergence of the results is consistent with the utility of these integrative approaches in the identification of the shared genetic and biological pathway information associated with COPD and its comorbid conditions.

In conclusion, we have integrated different levels of biological information (i.e., genes that are targeted by spatially constrained COPD-associated eQTLs, lung-specific GRN, LSPPIN, and all GWAS SNP-trait associations) to identify target genes, associated with COPD-associated eQTLs, that may interact to connect COPD to co-occurring conditions. Collectively, these results provide multiple new avenues for future investigation of the underlying biology and diverse clinical presentations of COPD. Empirical confirmation of the connections will suggest potential therapeutic COPD markers for follow-up patient stratification.

 **Life Science Alliance**

# Materials and Methods

### Hi–C data processing

Hi-C chromatin interaction libraries specific to primary lung cells (Table S1) were downloaded from GEO database (https://www.ncbi.nlm.nih.gov/geo/, accessions: GSM2322544 and GSM2322545) and analysed using the Juicer pipeline (v1.5) (Durand et al, 2016) to generate Hi-C libraries. The pipeline included BWA (v0.7.15) alignment of paired-end reads onto the hg38 (GRCh38; release 75) reference genome, merging paired-end read alignments and removing chimeric, unmapped and duplicated reads. We refer to the remaining read pairs as "contacts." Only Hi-C libraries that contain >90% alignable unique read pairs, and >50% unique contacts (<40% duplication rate) within the total sequenced read pairs were included in the analysis. Files containing cleaned Hi-C contacts (i.e. *_merged_nodups.txt files) were processed to obtain Hi-C chromatin interaction libraries in the following format: read name, str1, chr1, pos1, frag1 mapq1, str2, chr2, pos2, frag2, mapq2 (str, strand; chr, chromosome; pos, position; frag, restriction site fragment; mapq, mapping quality score; 1 and 2 correspond to read ends in a pair). Reads where both ends had a mapq ≥30 were included in the final library. Hi-C chromatin interactions represent all captured pairs of interacting restriction fragments in the genome and were used by CoDeS3D to identify putative regulatory interactions between SNPs and genes.

### Identification of SNPs associated with COPD, CAD, and UD

SNPs associated ($P < 5 \times 10^{-8}$) with COPD (n = 263), CAD (n = 804) and UD (n = 932) were downloaded from the GWAS Catalog (www.ebi.ac.uk/gwas/; 09/06/2021 and 11/04/2022; Table S2).

### Identification of spatial regulatory interactions

The CoDeS3D (Fadason et al, 2018) pipeline was used to identify genes that spatially interact with putative regulatory regions tagged by SNPs (Fig 1A). Briefly, the human genome build hg38 (GRCh38; release 75) was fragmented in silico at HindIII sites (ÁAGCTT), the restriction enzyme that was used in the preparation of the lung-specific Hi-C libraries (Schmitt et al, 2016). Disease associated SNP rsID numbers were cross-checked with the GTEx v8 lung eQTL database (GTEx Consortium, 2020) and restriction fragments that were tagged by the COPD associated SNPs were identified. Using lung-specific Hi-C libraries (Table S1), CoDeS3D identified the restriction fragments that were captured interacting with the SNP-tagged restriction HindIII fragments. Interacting fragments that overlapped annotated genes (GENCODE transcript model version 26) were identified. The resulting SNP-gene pairs were used to query the GTEx v8 lung eQTL database (GTEx Consortium, 2020) to identify cis- and trans-acting eQTLs (i.e. genes, whose expression levels are associated with the SNP identity). Finally, significant lung-specific eQTL–gene interactions were identified using the Benjamini–Hochberg (BH) FDR correction to adjust the eQTL *P*-values (FDR < 0.05).

CoDeS3D was used to build a lung-specific gene regulatory network (Fig S2). All SNPs (MAF ≥ 0.05, n = 43,174,097) present within

GTEx lung-specific eQTL database (GTEx Consortium, 2020) were used to identify all significant (BH, FDR < 0.05) spatially constrained lung-specific eQTL–gene interactions (the lung-specific GRN). Multiple correction testing was performed across all interactions within individual chromosomes.

The lung-specific GRN was mined using the COPD-associated SNPs (n = 263; $P < 5 \times 10^{-8}$; GWAS Catalog; 09/06/2021) to identify all COPD-associated significant (BH, FDR < 0.05) lung-specific eQTL–gene interactions (COPD-associated GRN). This was repeated for the CAD-associated (n = 651; $P < 5 \times 10^{-8}$; GWAS Catalog; 11/05/2022) and UD-associated (n = 152; $P < 5 \times 10^{-8}$; GWAS Catalog; 11/05/2022) SNPs (CAD- and UD-associated GRNs, respectively).

### Functional annotation of eQTL SNPs associated with COPD

The COPD-associated eQTLs were annotated using the wANNOVAR tool (Chang & Wang, 2012) (http://wannovar.wglab.org/, 09/06/2021) to obtain information about the locus they tagged. All genomic positions and SNP annotations were obtained for human genome reference build hg38 (GRCh38) release 75.

### Construction of lung-specific PPIN

The STRING (Szklarczyk et al, 2019) PPI database (version 11.5, https://string-db.org/, 15/03/2022) was downloaded and queried (STRING API) to identify potential PPIs (combined score ≥0.7). A lung-specific PPI network (LSPPIN) was constructed by filtering the STRING PPI network for the proteins encoded by the genes that were affected by eQTLs within the lung-specific GRN (Fig S3). Ensembl protein identifiers were mapped to Ensembl gene identifiers using EnsDb.Hsapiens.v86 R package. The LSPPIN represents a subnetwork of the entire STRING PPI network, in which a protein/node is only present if the encoding gene is associated with a spatially constrained eQTL within lung tissue. The size of each node depends on the protein expression levels (no missing values, TPM > 0.1 and ≥ 6 reads in a minimum of 20% of tested samples) within the GTEx v8 lung database (GTEx Consortium, 2020). The resulting LSPPIN contained 210,192 PPIs between 10,188 unique proteins. To build the COPD-specific LSPPIN, only interactions between genes targeted by COPD-associated eQTLs were extracted from the LSPPIN (Table S5).

### Identification of potential co-occurring conditions

The multimorbid3D pipeline was used to identify traits that were co-occurring with COPD (Fig 1B) (Zaied et al, 2022 *Preprint*). At "level 0" within the LSPPIN, we first identify the proteins encoded by the genes targeted by the COPD-associated eQTLs. At "level 1," the proteins interacting with the "level 0" proteins were identified. This process is iterated so that at each level, the proteins interacting with the proteins at the level minus 1 were identified. Next, we identify all eQTLs associated with the genes encoding the proteins at each level. The GWAS Catalog (www.ebi.ac.uk/gwas/, v1.0.2, 30/03/2022) was queried to identify traits that were enriched for the eQTLs from each level. The hypergeometric distribution test was used to identify significant enrichment of traits at each level within the GWAS Catalog traits (n = 17,841). Finally, significantly enriched

traits were identified using the BH FDR correction to adjust the *P*-values (FDR ≤ 0.05; Tables S6 and S7). The same pipeline was also used to identify the co-occurring conditions for CAD and UD (Table S8).

### GO enrichment and pathway analyses

GO enrichment and pathway analyses were performed using the g: GOSt module of the g:Profiler tool (Raudvere et al, 2019) (Tables S4 and S5). Enrichment was tested for within the biological process, molecular function and cellular component GO terms. All known human genes were chosen as the statistical domain scope. The significance level was determined using the BH algorithm (FDR < 0.05). The Kyoto Encyclopedia of Genes and Genomes database (Kanehisa & Goto, 2000) was used to identify impacted biological pathways.

### Bootstrapping analysis

Bootstrapping analysis (n = 1,000 iterations) was performed to test the specificity of the identified genes to COPD. At each bootstrap iteration, the same number of SNPs as were present in the tested condition were randomly selected from the GWAS Catalog (www.ebi.ac.uk/gwas/; 20/09/2022) and run through the CoDeS3D pipeline. The *P*-value was calculated as the number of the occurrences of the identified COPD-associated genes in the iterations divided by 1,000.

## Data Availability

Data access was approved by the dbGaP (https://www.ncbi.nlm.nih.gov/gap/) Data Access Committee(s) for total RNA-seq and WGS datasets across GTEx v8 tissues (project #22937: "Untangling the genetics of disease multimorbidity," accession: phs000424.v8.p2) (Aguet et al, 2019 *Preprint*) (Table S1). Data analysis and visualisation were performed in Python (version 3.6.9) using miniconda (version 4.8.4), and R (version 4.0.2) through RStudio (version 1.2.5033). All datasets and software used in the analysis are listed in Table S1.

All findings, scripts and a reproducibility report are available https://github.com/Genome3d/genetic_regulation_in_COPD.

Juicer is available at https://github.com/aidenlab/juicer.

CoDeS3D is available at https://github.com/Genome3d/codes3d-v2.

Multimorbid3D is available at https://github.com/Genome3d/multimorbid3D.

## Supplementary Information

## Acknowledgements

The authors would like to thank the Genomics and Systems Biology Group (Liggins Institute, University of Auckland) for useful discussions. The authors wish to acknowledge the use of New Zealand eScience Infrastructure (NeSI, https://www.nesi.org.nz) high-performance computing facilities, consulting support and/or training services as part of this research. New Zealand's national facilities are provided by NeSI and funded jointly by NeSI's collaborator institutions and through the Ministry of Business, Innovation & Employment's Research Infrastructure programme. This work was funded by the Dines Family Charitable Trust. The Genotype-Tissue Expression (GTEx) Project was supported by the Common Fund of the Office of the Director of the National Institutes of Health, and by NCI, NHGRI, NHLBI, NIDA, NIMH, and NINDS.

## Author Contributions

E Golovina: data curation, formal analysis, investigation, methodology, and writing—original draft.

T Fadason: resources, software, investigation, and writing—review and editing.

RK Jaros: resources, data curation, formal analysis, and writing—review and editing.

H Kumar: conceptualization and writing—review and editing.

J John: conceptualization and writing—review and editing.

K Burrowes: conceptualization and writing—review and editing.

M Tawhai: conceptualization, funding acquisition, and writing—review and editing.

JM O'Sullivan: conceptualization, resources, formal analysis, supervision, funding acquisition, project administration, and writing—original draft, review, and editing.

## Conflict of Interest Statement

The authors declare that they have no conflict of interest.

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
