## [Reviewer comments · Life Science Alliance]

Life Science Alliance

De novo discovery of traits co-occurring with chronic obstructive pulmonary disease

Evgeniia Golovina, Tayaza Fadason, Rachel Jaros, Haribalan Kumar, Joyce John, Kelly Burrowes, Merryn Tawhai, and Justin O'Sullivan

DOI: <https://doi.org/10.26508/lsa.202201609>

Corresponding author(s): *Justin O'Sullivan, The University of Auckland*

Review Timeline:

Submission Date:	2022-07-15
Editorial Decision:	2022-08-25
Revision Received:	2022-11-03
Editorial Decision:	2022-11-22
Revision Received:	2022-12-06
Accepted:	2022-12-08

Scientific Editor: Novella Guidi

Transaction Report:

August 25, 2022

Re: Life Science Alliance manuscript #LSA-2022-01609-T

Dr. Justin M O'Sullivan
University of Auckland
Liggins Institute
Private Bag 92019
Auckland Mail Center
Auckland 1023
New Zealand

Dear Dr. O'Sullivan,

Thank you for submitting your manuscript entitled "De novo discovery of traits co-occurring with chronic obstructive pulmonary disease" to Life Science Alliance. The manuscript was assessed by expert reviewers, whose comments are appended to this letter. We invite you to submit a revised manuscript addressing the Reviewer comments.

Thank you for this interesting contribution to Life Science Alliance. We are looking forward to receiving your revised manuscript.

Sincerely,

B. MANUSCRIPT ORGANIZATION AND FORMATTING:

Reviewer #1 (Comments to the Authors (Required)):

Golivina et al. present results of a multi-omic network analysis for discovery of traits co-occurring with COPD. Their analysis combines multiple sources of publicly available data - COPD GWAS results, Hi-C maps from bulk lung samples, eQTLs from lung tissue in the GTEx project, and gene interaction networks from STRING. Based on this analysis the authors conclude that "COPD-associated SNPs can control genes involved in the regulation of lung or pulmonary function, asthma, brain region volumes, cortical surface area, depressed affect, neuroticism, Parkinson's disease, white matter microstructure and smoking behaviour." The analysis is ambitious and shows sophistication with regard to the considering of gene regulation and the use of multiple datasets to try and construct reasonable network representations of gene regulatory networks in the lung. The handling of COPD and comorbidities could benefit from more sophistication in some areas, and some of the conclusions seem too strong given limitations of the analysis approach. The authors commendably have made much of their data analysis available on github, and there is a reproducibility document. Some key details of their analysis are however still not described well in the manuscript and as such it is not very easy to make any substantive determination on the validity of the results arising from these complex and ambitious analysis. The rationale and justifications for linking comorbidities to COPD is not very well stated, and the description of statistical significance testing is insufficient in many instances. Specific comments are included below:

- 1) The authors cite Zhou et al. 2013 to state that the heritability of COPD in smokers can be as high as 60%. It is not clear what this is based on as most of the data from the paper indicates a heritability for COPD and associated phenotypes in the range of 30-40% which is more in line with previous family-based studies.
- 2) This statement made by the authors - "This variation in heritability suggests that the presence of two or more conditions (hereafter defined as co-occurring) can increase the risk of mortality in COPD patients." - does not seem logically connected. It is not clear to me how studies of the estimated genetic component of COPD shed light on co-morbidities and their relationship to mortality risk in COPD.
- 3) When citing GWAS studies of COPD and "its individual co-occurring conditions", the authors cite Kim et al. 2021 and Zhu et al. 2019, but this leaves out many relevant GWAS studies, most notable the studies by Sakornsakolpat and Shrine in Nature Genetics 2019 which are by far the largest and most definitive GWAS of COPD and lung function performed to date.
- 4) There are references to FDR throughout the manuscript, but it is often difficult to ascertain what specific statistical tests have been performed in each case. For example, this description of bootstrapping in the Methods does not include sufficient detail to be clear on the analysis to which this applies and the specifics of how the bootstrapping was performed - "Bootstrapping (n=1000 iterations) was performed to test the specificity of the identified genes, pathways and traits to COPD. The p-value was calculated as the number of the occurrences of COPD-associated genes in the iterations divided by 1000."
- 5) What are levels 0-4 in the COPD LSPPIN? Please provide a clearer description of how these levels are determined and what their interpretation is.
- 6) I did not understand this sentence or the authors meaning of "total" eQTLs - "We contend that the total eQTLs we identified, as impacting on COPD and its co-occurring traits, represent the 'total' genetic burden that contributes to an individual's risk of developing COPD and its diverse clinical presentations."
- 7) It seems that the only network analyzed was a lung-specific network, but from the analysis of this network biological overlap was demonstrated between COPD and phenotypes such as brain volume and Parkinson's disease. It seems counterintuitive to suggest that a lung-specific gene regulatory network would have enough overlap with gene regulatory mechanisms in the brain for such an overlap to be identified. It would seem important to carefully consider alternative explanations in cases like this - it is possible that there is some shared biology in the lung and brain that perhaps comes through in this analysis, but it also seems possible that the enrichment tests on which this conclusion is based are biased. In general bias is difficult to avoid in pathway enrichment tests and certain methods are more susceptible to this than others. Some comment on gProfiler and the g:GOST module would be appropriate here - has it been benchmarked against other methods with respect to its statistical performance?
- 8) In relation to 7), in the description of the pathway enrichment performed with gProfiler the authors used all annotated human genes as the "statistical domain scope." Typically the proper background set of genes for enrichment analysis should be the set of genes included in a given analysis, since genes not included in the analysis have no chance of showing up as significant results.
- 9) In the Discussion, the authors state that "COPD patients may exhibit progressive structural impairments in both the grey and white matter, along with impaired levels of lung function. In addition to the trait associations, the results of our study also provide evidence for the putative genetic and biological connections between these traits." This is an interesting observation but the assessment of this strength of evidence seems optimistic. There does seem to be some epidemiologic association between

COPD and Alzheimer's disease in observation data (<https://bmjopenrespres.bmj.com/content/8/1/e000759>), but what would be most helpful would be some evidence to suggest that specific biological entities are involved in a causal way in both diseases - in that light, some discussion of what sort of causal inferences can and cannot be drawn from this approach would be important, and should be placed in the context of other relevant methods such as Mendelian randomization or mediation analysis, which are simpler approaches whose causal inference is nonetheless dependent on strong assumptions.

10) The authors are to be commended for creating the github resource and making the reproducibility document available for review. This was only partially reviewed by this reviewer due to time constraints. Some suggestions for the reproducibility would be a) provide more detail and basic sanity checks and QC on the source datasets and b) on some cases the figures do not seem to be reflecting actual computations but are instead relying on numbers that were pre-computed elsewhere so in this sense reproducibility hasn't really been demonstrated. For example the piechart for eQTL and non-eQTL SNPs in section 1.1 is generated with this code:

```
lung.map.genome <- data.frame(  
snps = rep(c("eQTL", "non-eQTL"),  
number = c(740028, (43174097-740028)),  
percentage = c(round((740028/43174097)*100, 2),  
round(((43174097-740028)/43174097)*100, 2)))
```

which seems to be simply plugging in numbers obtained elsewhere rather than computing them directly from the data.

11) The Discussion should include some comparison to other eQTL network-based analysis of COPD, such as <https://journals.plos.org/ploscompbiol/article?id=10.1371/journal.pcbi.1005033>.

Reviewer #2 (Comments to the Authors (Required)):

1.- A short summary of the paper, including description of the advance offered to the field.

Chronic obstructive pulmonary disease (COPD) is a complex disease, with environmental but also genetic factors as well as known to co-occur with other conditions. Golovina et al present here a very interesting method to study and better characterise the diversity of this disease: they started by expanding common variants from GWAS to lung specific eQTLs and integrating this information with a lung-specific gene regulatory network and protein protein interaction network. Together with common variation from all traits contained in the GWAS catalog, they are able to study the biological mechanisms behind co-occurrence, reaching a better understanding of COPD, needed for improvements in the clinics (better stratification and better treatments).

2.- For each main point of the paper, please indicate if the data are strongly supportive. If not, explicitly state the additional experiments essential to support the claims made and the timeframe that these would require.

I think that the method itself is potentially interesting and a valid way to study disease co-occurrence but there are some points where the presented evidence to unravel biological mechanisms are weak and should be revised. These are my main objections:

A) I'm not convinced with the network analysis performed in the results section headed "COPD-associated genes are enriched for diverse biological processes in the lung" in page 5 and the associations it leads to. This will be the PPI interaction network from STRING after several filtering steps, leading to 107 nodes. First, I would like to know if the network was considered as directed or undirected, this is relevant for community detection and also to understand the meaning of the duplicated edges that you can see in figure 2. But my main concern lies in the size of the clusters: 6 of them only have 2 nodes, 2 have 4 nodes and the remaining one 3 nodes most of them with poor connection). This fact makes me think that the communities detected here using Louvain were subnetworks, disconnected from the very beginning and not a result of the clustering algorithm per se. I made a quick STRING query myself and it looks like the case for several clusters (from table 6, all clusters with the exception of cluster 3). Another reason why the size of the clusters is worrying is the pathway enrichment analysis that follows it: having one gene associated to one particular pathway (as it happens with "TGF-beta signalling pathway" in cluster 1, "Sulphur relay system" and "Folate biosynthesis" in cluster 6 or "Axon guidance" and "Regulation of actin cytoskeleton" in cluster 9, according to table S5), might be statistically significant but not so relevant from a biological point of view. This is even more weak if not all the proteins in the cluster have a connection within each other. And sadly this is the case for the two examples from figure 4 (A & B): they are based on two of these biological pathways poorly sustained by only one gen each, MOCS2 (linked with both sulfur relay system and folate biosynthesis and interacting only with SGF29 and not the other 2 genes in the cluster) and SSH2 for axon guidance and actin cytoskeleton. I recommend to completely exclude the clustering and subsequent enrichment analysis and find another way of showing the results, maybe all connected components (all subnetworks) together with pathway enrichment of all nodes and not just the little clusters. With the evidence that you are showing now (clusters and pathway enrichment), stating that "MSL1, MOCS2, NUPR1 and SGF29 were involved in the sulfur relay system and folate biosynthesis pathways" is misleading, as only MOCS2 form part of both pathways and it is known to interact only with SGF29 (the quoted sentence that can be found in page 8 of the manuscript)

B) When I was examining figure 4B during the previous point explanation I found that neither SSH2 nor TESK2 interact with BIN1 at least at protein level as it is stated in both the main text and the legend of figure 4. This was manually checked using STRING. Please revise and amend accordingly because this interaction is key in the association of the subsequent diseases.

C) In the discussion section there is a paragraph where the authors underline the connection with Impaired lung function exemplified by several traits from the co-occurrence study and insulin resistance and immune pathways from the pathways

enrichment analysis. Taking into account my previous objections (point A) I would like to see a stronger evidence of this connexion, for example, are the proteins that lead to the enrichment results at the different levels between COPD and the exemplified impaired lung function traits (FVC and FEV1) related with insulin resistance or immune related diseases?

D) The authors state in several points of the manuscript that they have defined the biological connection between COPD and other conditions (for example in the discussion and the last part of the abstract) but based on my previous objections I think that the established connections are a bit weak. There are several actions that can be accomplished to add strength to the different points: One possibility could be to recapitulate the conclusions concerning co occurrence using protein expression data in patients, (Expression Atlas could help to fetch the needed data, <https://www.ebi.ac.uk/gxa/home>). Other sources of evidence could be therapeutic targets, do therapies for COPD or the other 39 conditions share therapeutic targets and if so, which biological processes are the targets part of?

3.- Lastly, indicate any additional issues you feel should be addressed (text changes, data presentation, statistics etc.).

A) Tables 2 & 3: the data presented in the first paragraph of the results section (page 3) concerning the genomic regions is not easy to reproduce, as the information is contained in a column in table 2 that then needs to be merged with table 3, where the eQTLs are. I suggest either adding an extra column in table 2 to state if a given SNP is an eQTL or adding an extra column in table 3 for the genetic region annotation or both things.

B) The genomic annotation in figure 1B does not correspond with the one provided in table S2, in some terms this is not a problem as they are close enough but I was unable to find anything similar to "exonic". I suggest clarifying this in the legend or in the tables.

C) Material and methods in the section "Gene Ontology enrichment and pathway analysis", supplemental tables S8 & S9 do not exist in the material I have access to suggesting a typo.

D) The number of traits extracted from the GWAS catalog and used in the Multimorbid 3D pipeline could be a nice addition to the material and methods and maybe to the main text as well.

E) Results from enrichment analysis: are the p-values corrected for multiple testing? It is clear for table 4 (no correction) but not so much for table 6, the text suggests FDR calculations (page 5 "Pathway analysis of the nine highly connected modules identified seven that were enriched (FDR < 0.05)") but the table does not show this.

F) I would like an extra supplementary table showing which eQTLs (mapped to which genes etc etc) as well as which proteins at the different levels are co occurring with the significant 39 GWAS traits. This will help to visualise which genes are responsible for the different overlaps, if some of them are never overlapping (COPD-unique), etc. The total number of genes on each level should be also noted to help with the interpretation

G) In the methods there is a bootstrap method described but it is not specified where it was implemented. It will be useful to clarify that information.

H) Figure4: I think that showing the real connection between the proteins with edges joining the boxes of the heatmap will highly improve the information given here, instead of just stating it in the text without specifying which proteins are involved.

H) Typos: page 2 "Table 3S", page 13 section "Identification of potential co-occurring" also inside the section the first mentioned table should be 6 instead of 7.

Reviewer 1:

1. The authors cite Zhou et al. 2013 to state that the heritability of COPD in smokers can be as high as 60%. It is not clear what this is based on as most of the data from the paper indicates a heritability for COPD and associated phenotypes in the range of 30-40% which is more in line with previous family-based studies.

➤ We thank the referee for pointing out this error. We have modified the statement as follows:

“The estimated genetic heritability of COPD ranges from 20% to 40% (Stolz 2020; Gim et al. 2020; Zhou et al. 2013).”

2. This statement made by the authors - "This variation in heritability suggests that the presence of two or more conditions (hereafter defined as co-occurring) can increase the risk of mortality in COPD patients." - does not seem logically connected. It is not clear to me how studies of the estimated genetic component of COPD shed light on co-morbidities and their relationship to mortality risk in COPD.

➤ We agree with the reviewer and have deleted this statement. We have replaced it with the following

“Co-occurring conditions are widely recognised as impacting on COPD patient outcomes (Cavallès et al. 2013; Burke and Wilkinson 2021). As such, a better understanding of COPD co-occurring conditions is essential to enable effective management, therapeutic optimization and reduce the costs of managing COPD patients (Mannino et al. 2015).”

3. When citing GWAS studies of COPD and "its individual co-occurring conditions", the authors cite Kim et al. 2021 and Zhu et al. 2019, but this leaves out many relevant

GWAS studies, most notable the studies by Sakornsakolpat and Shrine in Nature Genetics 2019 which are by far the largest and most definitive GWAS of COPD and lung function performed to date.

➤ We apologise for this omission and have included these citations:

“To date, genome-wide association studies (GWASs) have identified common single nucleotide polymorphisms (SNPs) that are associated with COPD, or its individual co-occurring conditions (Kim et al. 2021; Zhu et al. 2019; Sakornsakolpat et al. 2019; Shrine et al. 2019).”

4. There are references to FDR throughout the manuscript, but it is often difficult to ascertain what specific statistical tests have been performed in each case. For example, this description of bootstrapping in the Methods does not include sufficient detail to be clear on the analysis to which this applies and the specifics of how the bootstrapping was performed - "Bootstrapping (n=1000 iterations) was performed to test the specificity of the identified genes, pathways and traits to COPD. The p-value was calculated as the number of the occurrences of COPD-associated genes in the iterations divided by 1000."

➤ We have modified the methods to read:

“Bootstrapping analysis (n=1000 iterations) was performed to test the specificity of the identified genes to COPD. At each bootstrap iteration the same number of SNPs as were present in the tested condition were randomly selected from the GWAS Catalog (www.ebi.ac.uk/gwas/; 20/09/2022) and run through the CoDeS3D pipeline. The p-value was calculated as the number of the occurrences of the identified COPD-associated genes in the iterations divided by 1000.”

5. What are levels 0-4 in the COPD LSPPIN? Please provide a clearer description of how these levels are determined and what their interpretation is.

➤ We have modified the text to read:

“The Multimorbid3D pipeline was used to identify traits that were co-occurring with COPD (Fig. 1B) (Zaied et al. 2022). At “level 0” within the LSPPIN, we first identify the proteins encoded by the genes targeted by the COPD-associated eQTLs. At “level 1”, the proteins interacting with the “level 0” proteins were identified. This process is iterated so that at each level, the proteins interacting with the proteins at the level minus 1 were identified. Next, we identify all eQTLs associated with the genes encoding the proteins at each level. The GWAS Catalog (www.ebi.ac.uk/gwas/, v1.0.2, 30/03/2022) was queried to identify traits that were enriched for the eQTLs from each level. The hypergeometric distribution test was used to identify significant enrichment of traits at each level within the GWAS Catalog traits (n=17,841). Finally, significantly enriched traits were identified using the BH FDR correction to adjust the p values ($FDR \leq 0.05$; Supplemental Tables S6 & S7). The same pipeline was also

used to identify the co-occurring conditions for CAD and UD (Supplemental Table S8).”

6. I did not understand this sentence or the authors meaning of "total" eQTLs - "We contend that the total eQTLs we identified, as impacting on COPD and its co-occurring traits, represent the 'total' genetic burden that contributes to an individual's risk of developing COPD and its diverse clinical presentations."

➤ By saying “total”, we mean the complete set of eQTLs associated with COPD and its co-occurring traits at the population level. At the personal level, each individual will have a subset of the total SNP set, and thus eQTLs, gene and protein effects associated with a risk of particular subset of the co-occurring traits and COPD.

➤ We have modified the statement as follows:

“We contend that the eQTLs we identified, as impacting on COPD and its co-occurring traits, represent the population-based genetic burden that may contribute to an individual’s risk of developing COPD and its diverse clinical presentations.”

7. It seems that the only network analyzed was a lung-specific network, but from the analysis of this network biological overlap was demonstrated between COPD and phenotypes such as brain volume and Parkinson's disease. It seems counterintuitive to suggest that a lung-specific gene regulatory network would have enough overlap with gene regulatory mechanisms in the brain for such an overlap to be identified. It would seem important to carefully consider alternative explanations in cases like this

➤ The overlap we have observed is driven by pleiotropic genes that have regulatory contributions from eQTLs in the lung, the SNPs of which have been associated with brain-related phenotypes (e.g. brain volume, Parkinson’s disease) within the GWAS Catalog. While we agree that it appears counterintuitive, the effects of regulatory elements are combinatorial and there is potential for shared gene regulatory mechanisms between the lung and the brain. For example, loss of nuclear factor I site-specific transcription factors results in apparent arrest of fetal lung maturation and development of the forebrain and hindbrain in mice (Steele-Perkins et al. 2005). There is no reason to assume that this regulatory overlap only occurs early in development and that it cannot be realised in the adult lung or brain. This does not rule out the possibility that there is Functional Two-Way Crosstalk Between Brain and Lung that impacts indirectly on the regulatory network (Li et al. 2022). However, as we stress in our manuscript, these results are putative until empirically confirmed.

8. - it is possible that there is some shared biology in the lung and brain that perhaps comes through in this analysis, but it also seems possible that the enrichment tests on which this conclusion is based are biased. In general bias is difficult to avoid in pathway enrichment tests and certain methods are more susceptible to this than others. Some comment on gProfiler and the g:GOST module would be appropriate here - has it been benchmarked against other methods with respect to its statistical performance?

➤ We have been careful to try to minimise bias in our analysis using g:Profiler. In g:Profiler, the p value of the enrichment of a pathway is computed using a Fisher's exact test before the multiple-test correction (e.g. BH-FDR) is applied (Reimand et al. 2019). By default, g:Profiler uses all "annotated" genes as a statistical domain scope. However, in our analysis we analysed all known human genes not just annotated. Measuring only a subset of all genes can lead to "inappropriate inflation of p values and false-positive results". Therefore, to avoid this bias, we changed the statistical domain scope to all "known" genes. This is stated in the methods section: "All known human genes were chosen as the statistical domain scope."

9. In relation to 7), in the description of the pathway enrichment performed with gProfiler the authors used all annotated human genes as the "statistical domain scope." Typically the proper background set of genes for enrichment analysis should be the set of genes included in a given analysis, since genes not included in the analysis have no chance of showing up as significant results.

➤ We have run g:Profiler with default statistical domain scope – "annotated" (only the genes that have at least one annotation) and the domain scope using all "known" human genes (28 September 2022). The results were similar, except one of the previously unannotated clusters became annotated with "Phagosome" pathway:

Cluster 1: TGF-beta signaling pathway (ADAMTSL3, THSD4)

Cluster 2: Regulation of actin cytoskeleton, Focal adhesion (BCAR1, ITGA1)

Cluster 3: Neuroactive ligand-receptor interaction, Cholinergic synapse (CHRNA3, CHRNA5, FAM13A, IREB2)

Cluster 4: Unknown (DMPK, SIX5)

Cluster 5: Immune-related pathways (HLA-C, HLA-DQB1)

Cluster 6: Sulfur relay system, Folate biosynthesis (MOCS2, MSL1, NUPR1, SGF29)

Cluster 7: Insulin resistance, Insulin signaling pathway (PPP1CB, PPP1R18, PPP1R3B)

Cluster 8: Phagosome (SCGB1A1, SFTPD)

Cluster 9: Axon guidance, Regulation of actin cytoskeleton (SSH2, TESK)

As suggested by the referee we have only shown results for the "all known human genes" analysis.

➤ We have also removed references to pathway enrichment analyses on clusters of genes (Figure 2). We now present all subnetworks and performed pathway enrichment on all 107 COPD-associated genes (Figure 2, Supplementary table S5). We have updated the Methods and Results sections to reflect this change.

10. In the Discussion, the authors state that "COPD patients may exhibit progressive structural impairments in both the grey and white matter, along with impaired levels of lung function. In addition to the trait associations, the results of our study also provide evidence for the putative genetic and biological connections between these traits." This is an interesting observation but the assessment of this strength of

evidence seems optimistic. There does seem to be some epidemiologic association between COPD and Alzheimer's disease in observation data (<https://bmjopenrespres.bmj.com/content/8/1/e000759>), but what would be most helpful would be some evidence to suggest that specific biological entities are involved in a causal way in both diseases - in that light, some discussion of what sort of causal inferences can and cannot be drawn from this approach would be important, and should be placed in the context of other relevant methods such as Mendelian randomization or mediation analysis, which are simpler approaches whose causal inference is nonetheless dependent on strong assumptions.

➤ We have modified the Discussion by including the following:

“In addition to the trait associations, the results of our study provide putative evidence for the existence of genetic and biological connections between these traits (Higbee et al. 2021). In so doing our results stop short of providing causal evidence for these putative connections and two sample or multivariable Mendelian randomization (e.g. Transcriptome-Wide Mendelian Randomization) or mediation analyses are required to clarify which if any of the interactions are causal (Russ et al.; Porcu et al. 2019).”

11. The authors are to be commended for creating the github resource and making the reproducibility document available for review. This was only partially reviewed by this reviewer due to time constraints. Some suggestions for the reproducibility would be a) provide more detail and basic sanity checks and QC on the source datasets and b) on some cases the figures do not seem to be reflecting actual computations but are instead relying on numbers that were pre-computed elsewhere so in this sense reproducibility hasn't really been demonstrated. For example the piechart for eQTL and non-eQTL SNPs in section 1.1 is generated with this code:

```
lung.map.genome <- data.frame(  
  snps = rep(c("eQTL","non-eQTL"),  
    number = c(740028, (43174097-740028)),  
    percentage = c(round((740028/43174097)*100, 2),  
      round(((43174097-740028)/43174097)*100, 2)))
```

which seems to be simply plugging in numbers obtained elsewhere rather than computing them directly from the data.

➤ We thank the referee for their comments on this section of our work. We have modified the reproducibility report to compute the numbers directly from the data.

12. The Discussion should include some comparison to other eQTL network-based analysis of COPD, such as <https://journals.plos.org/ploscompbiol/article?id=10.1371/journal.pcbi.1005033>.

➤ The CONDOR method utilizes eQTL but is different from our analyses, using a more relaxed FDR (10% vs 5%, respectively). Moreover, CONDOR does not incorporate

tissue-specific spatial genome organization information (e.g. lung-specific chromatin interactions [Hi-C]) into the analysis. Finally, CONDOR represents the eQTL-gene associations as a bipartite network consisting of two classes of nodes – eQTLs and genes – with edges from eQTLs to the genes. We contend that incorporating protein-protein interactions into the analysis helps understand the complex gene-gene interactions within a cell or between different cells. Despite these differences, there is convergence about the pathways that are associated with COPD. Therefore, we have included the following in our discussion:

- “Integrative network-based methods have been used to explore complex SNP-gene interactions that can impact functional biological pathways and lead to complex phenotypes (Platig et al. 2016; Chen et al. 2021; Zhu et al. 2021). The CONDOR algorithm was applied to study eQTLs in COPD (Platig et al. 2016). In brief, CONDOR built a bipartite network linking eQTLs and genes in 52 communities. 30 SNPs associated with COPD (Cho et al. 2014) mapped to three of these communities. Notably, despite only one gene (KANS1) being identified in Platig et al. 2016 and our current study, the loci involved were enriched in conserved biological processes including Alzheimer’s, Parkinson’s and Asthma and immune responses. This is despite fundamental difference in the methods (e.g. eQTLs vs spatially constrained eQTLs, and the inclusion of the expanded protein network as additional information in Multimorbid3D) and updates in the GWAS catalog that occurred between 2016 and 2022. As such, the apparent convergence of the results is consistent with the utility of these integrative approaches in the identification of the shared genetic and biological pathway information associated with COPD and its comorbid conditions.”

Reviewer 2:

1. I'm not convinced with the network analysis performed in the results section headed "COPD-associated genes are enriched for diverse biological processes in the lung" in page 5 and the associations it leads to. This will be the PPI interaction network from STRING after several filtering steps, leading to 107 nodes. First, I would like to know if the network was considered as directed or undirected, this is relevant for community detection and also to understand the meaning of the duplicated edges that you can see in figure 2.
 - The network was considered as undirected. We apologize for the representation of duplicated edges in the original Figure 2. This was an error in our visualization. We have updated Figure 2 to illustrate the information categories that support the connection in STRING.
2. But my main concern lies in the size of the clusters: 6 of them only have 2 nodes, 2 have 4 nodes and the remaining one 3 nodes most of them with poor connection). This fact makes me think that the communities detected here using Louvain were subnetworks, disconnected from the very beginning and not a result of the clustering algorithm per se. I made a quick STRING query myself and it looks like the

case for several clusters (from table 6, all clusters with the exception of cluster 3). Another reason why the size of the clusters is worrying is the pathway enrichment analysis that follows it: having one gene associated to one particular pathway (as it happens with "TGF-beta signalling pathway" in cluster 1, "Sulphur relay system" and "Folate biosynthesis" in cluster 6 or "Axon guidance" and "Regulation of actin cytoskeleton" in cluster 9, according to table S5), might be statistically significant but not so relevant from a biological point of view. This is even more weak if not all the proteins in the cluster have a connection within each other. And sadly this is the case for the two examples from figure 4 (A & B): they are based on two of these biological pathways poorly sustained by only one gene each, MOCS2 (linked with both sulfur relay system and folate biosynthesis and interacting only with SGF29 and not the other 2 genes in the cluster) and SSH2 for axon guidance and actin cytoskeleton. I recommend to completely exclude the clustering and subsequent enrichment analysis and find another way of showing the results, maybe all connected components (all subnetworks) together with pathway enrichment of all nodes and not just the little clusters. With the evidence that you are showing now (clusters and pathway enrichment), stating that "MSL1, MOCS2, NUPR1 and SGF29 were involved in the sulfur relay system and folate biosynthesis pathways" is misleading, as only MOCS2 form part of both pathways and it is known to interact only with SGF29 (the quoted sentence that can be found in page 8 of the manuscript).

- We thank the reviewer for their comments. We have excluded clustering and subsequent pathway enrichment analysis on clusters of genes. Instead, we have shown all subnetworks and performed pathway enrichment on all 107 COPD-associated genes. We have updated Figure 2 and Supplementary table S5 and modified the Methods section to reflect this:

“The STRING (Szklarczyk et al. 2019) protein-protein interaction (PPI) database (version 11.5, <https://string-db.org/>, 15/03/2022) was downloaded and queried (STRING API) to identify potential protein-protein interactions (combined score ≥ 0.7). A lung-specific PPI network (LSPPIN) was constructed by filtering the STRING PPI network for the proteins encoded by the genes that were affected by eQTLs within the lung-specific GRN (Supplemental Fig. S3). Ensembl protein identifiers were mapped to Ensembl gene identifiers using EnsDb.Hsapiens.v86 R package. The LSPPIN represents a subnetwork of the entire STRING PPI network, in which a protein/node is only present if the encoding gene is associated with a spatially constrained eQTL within lung tissue. The size of each node depends on the protein expression levels (no missing values, TPM >0.1 and ≥ 6 reads in a minimum of 20% of tested samples) within the GTEx v8 lung database (GTEx Consortium 2020). The resulting LSPPIN contained 210,192 PPIs between 10,188 unique proteins. To build the COPD-specific LSPPIN, only interactions between genes targeted by COPD-associated eQTLs were extracted from the LSPPIN (Supplemental Table S5).”

- Also, we changed the statistical domain scope to all “known” genes. This is stated in the Methods section as follows:

“All known human genes were chosen as the statistical domain scope.”

- Additionally, we have modified the Results section as follows:

“Functional gene ontology enrichment analysis identified metabolic, behavioural, regulatory and protein modification processes (e.g. “phosphorus metabolic process”, “behavioral response to nicotine”, “regulation of postsynaptic membrane potential”, “protein acetylation” and “protein acylation”) as being significantly ($p < 0.05$) enriched within the 107 COPD-associated genes (Supplemental Table S4, Supplemental Fig. S4). These 107 genes encoded proteins that formed nine COPD-associated lung-specific protein-protein interaction subnetworks (Fig. 2). Pathway analysis of these 107 COPD-associated genes identified biological pathways that were enriched ($FDR < 0.05$) for regulation of actin cytoskeleton, insulin signaling and resistance, focal adhesion, phagosome, immune processes, infections and diseases, alcoholism, long-term depression (Supplemental Table S5).”

3. When I was examining figure 4B during the previous point explanation I found that neither SSH2 nor TESK2 interact with BIN1 at least at protein level as it is stated in both the main text and the legend of figure 4. This was manually checked using STRING. Please revise and amend accordingly because this interaction is key in the association of the subsequent diseases.

- Thank you for pointing out our error. BIN1 interacts with SSH2 through PPP3CC, so it is a level 2 not level 1 protein. We have modified Figure 4B to remove reference to Bin1.

4. In the discussion section there is a paragraph where the authors underline the connection with Impaired lung function exemplified by several traits from the co-occurrence study and insulin resistance and immune pathways from the pathways enrichment analysis. Taking into account my previous objections (point A) I would like to see a stronger evidence of this connexion, for example, are the proteins that lead to the enrichment results at the different levels between COPD and the exemplified impaired lung function traits (FVC and FEV1) related with insulin resistance or immune related diseases?

- We have modified the Discussion by including the following:

“Impaired lung function as measured by forced vital capacity (FVC) or forced expiratory volume in the first second (FEV1) has been previously associated with insulin resistance (Machado et al. 2018; Piazzolla et al. 2017; Sagun et al. 2015; Kim et al. 2021). The evidence that supports the existence of a relationship between COPD and insulin resistance is complex. Our independent identification that PPP1CB and

PPP1R3B are COPD-associated genes is notable because these genes are directly involved in controlling glycogen synthesis and glucose homeostasis in insulin signaling (Li et al. 2019) and have been linked to glycaemic traits (Niazi et al. 2019). We also identified several COPD-associated genes enriched for immune processes and immune-related diseases (e.g. HLA-C and HLA-DQB1 - antigen processing and presentation, HLA-DQB1 - type I diabetes mellitus, Influenza A, autoimmune thyroid disease) (Supplemental Table S5). This finding is consistent with observations that proteins of the major histocompatibility complex of classes I and II (HLA-I and HLA-II) have been identified as potential markers of progression of systemic and local inflammation in patients with COPD (Kubysheva et al. 2018). Specifically, an increase in the level of HLA-I and HLA-II molecules in the exhaled breath condensate as well as an elevated serum level of HLA-II is observed in COPD patients when compared to healthy volunteers (Kubysheva et al. 2018). Previous studies have also identified elevated levels of plasma transforming growth factor- β (TGF- β), an important regulator of lung and immune system development, in COPD patients compared to healthy controls (Mak et al. 2009; Verhamme et al. 2015). Despite the apparent correlative support between our results and previously published findings, the evidence for a role for the insulin resistance pathway and immune system in COPD remains putative until proven empirically.”

5. The authors state in several points of the manuscript that they have defined the biological connection between COPD and other conditions (for example in the discussion and the last part of the abstract) but based on my previous objections I think that the established connections are a bit weak. There are several actions that can be accomplished to add strength to the different points: One possibility could be to recapitulate the conclusions concerning co-occurrence using protein expression data in patients, (Expression Atlas could help to fetch the needed data, <https://www.ebi.ac.uk/gxa/home>). Other sources of evidence could be therapeutic targets, do therapies for COPD or the other 39 conditions share therapeutic targets and if so, which biological processes are the targets part of?
- We agree that the identified SNP-gene-trait associations are putative and require validation. Additional studies are required to confirm the causal relationships between COPD-associated spatial eQTLs and other conditions. Mendelian randomization analyses and the analyses outlined by the referee would be a beginning. However, mechanistic evidence will require extensive biochemical and biological experiments. Unfortunately, we lack sufficient experience to complete these experiments. We contend that our discovery analysis will inform the development and testing of hypotheses by researchers with the appropriate skills to falsify them. We have changed the conclusion of the manuscript to read:

“In conclusion, we have integrated different levels of biological information (i.e. genes that are targeted by spatially constrained COPD-associated eQTLs, lung-specific GRN, LSPPIN and all GWAS SNP-trait associations) to identify target genes, associated with COPD-associated eQTLs, that may interact to connect COPD to co-occurring conditions. Collectively, these results provide multiple new avenues for future investigation of the underlying biology and diverse clinical presentations of

COPD. Empirical confirmation of the connections will suggest potential therapeutic COPD markers for follow-up patient stratification.”

➤ We have also modified the discussion, in several other places, to reinforce the putative nature of our observations and the need for further empirical confirmation.

6. Lastly, indicate any additional issues you feel should be addressed (text changes, data presentation, statistics etc.).

1. Tables 2 & 3: the data presented in the first paragraph of the results section (page 3) concerning the genomic regions is not easy to reproduce, as the information is contained in a column in table 2 that then needs to be merged with table 3, where the eQTLs are. I suggest either adding an extra column in table 2 to state if a given SNP is an eQTL or adding an extra column in table 3 for the genetic region annotation or both things.

➤ We thank the referee for their comment. We have added an extra column “eQTL SNP” to Supplemental Table S2 highlighting if the GWAS SNP is an eQTL in the lung tissue.

2. The genomic annotation in figure 1B does not correspond with the one provided in table S2, in some terms this is not a problem as they are close enough but I was unable to find anything similar to "exonic". I suggest clarifying this in the legend or in the tables.

➤ We apologize for the confusion. The genomic annotation in Supplemental Figure S1B was generated by wANNOVAR. The information in Supplemental Table S2 contains the original information from GWAS Catalog. We have changed the results section to read:

“Approximately 96% of the identified eQTLs were located within non-coding genomic regions, with 66.02% and 18.45% of them being intronic and intergenic, respectively (Supplemental Fig. S1A and S1B [wANNOVAR annotation], Supplemental Table S2 [original GWAS Catalog annotation]).”

➤ We have included the functional annotation of the eQTL SNPs by the wANNOVAR tool as a separate column in the Supplemental Table S3.

➤ We have changed the legend to Supplemental figure S1 to read:

“Characteristics of COPD-associated lung-specific GRN. (A) Of 263 COPD-associated GWAS SNPs, 103 SNPs are involved in spatially constrained eQTL-gene interactions in the lung. (B) wANNOVAR annotations identified approximately 96% of the COPD-associated eQTLs are located within non-coding genomic regions (Supplemental Table S3).”

3. Material and methods in the section "Gene Ontology enrichment and pathway analysis", supplemental tables S8 & S9 do not exist in the material I have access to suggesting a typo.

➤ We have fixed the typos:

“Supplemental Table S8” → “Supplemental Table S4”

“Supplemental Table S9” → “Supplemental Table S5”

4. The number of traits extracted from the GWAS catalog and used in the Multimorbid 3D pipeline could be a nice addition to the material and methods and maybe to the main text as well.

➤ We have modified the text to include the following information in the Methods section:

“The hypergeometric distribution test was used to identify significant enrichment of traits at each level within the GWAS Catalog traits (n=17,841).”

5. Results from enrichment analysis: are the p-values corrected for multiple testing? It is clear for table 4 (no correction) but not so much for table 6, the text suggests FDR calculations (page 5 "Pathway analysis of the nine highly connected modules identified seven that were enriched (FDR < 0.05)") but the table does not show this.

➤ We have double checked and for both GO and pathway enrichment analyses, the FDR correction (FDR < 0.05) was applied.

6. I would like an extra supplementary table showing which eQTLs (mapped to which genes etc etc) as well as which proteins at the different levels are co-occurring with the significant 39 GWAS traits. This will help to visualise which genes are responsible for the different overlaps, if some of them are never overlapping (COPD-unique), etc. The total number of genes on each level should be also noted to help with the interpretation

➤ We have included Supplemental Table S7 – for identified significant trait-eQTL-gene associations (at different PPI levels) for COPD.

7. In the methods there is a bootstrap method described but it is not specified where it was implemented. It will be useful to clarify that information.

➤ We modified the text as follows:

“Bootstrapping analysis (n=1000 iterations) was performed to test the specificity of the identified genes to COPD. At each bootstrap iteration the same number of SNPs as were present in the tested condition were randomly selected from the GWAS Catalog (www.ebi.ac.uk/gwas/; 20/09/2022) and run through the CoDeS3D pipeline. The p-value was calculated as the number of the occurrences of the identified COPD-associated genes in the iterations divided by 1000.”

8. Figure4: I think that showing the real connection between the proteins with edges joining the boxes of the heatmap will highly improve the information given here, instead of just stating it in the text without specifying which proteins are involved.
- We thank the reviewer for their comment. We have plotted all the “level 0” and “level 1” genes within “NUPR1-MSL1-SGF29-MOCS2” and “TESK2-SSH2” PPI subnetworks in the Supplementary Figure S5. In Figure 4, we present only genes that have eQTLs (within the lung-specific GRN) and putative eQTL-trait associations from the GWAS Catalog.
9. Typos: page 2 "Table 3S", page 13 section "Identification of potential co-occurring" also inside the section the first mentioned table should be 6 instead of 7.
- We have fixed the typo “Table 3S” → “Table S3”.
 - We have mentioned “Supplemental Table S6” instead of “Supplemental Table S7”.

References:

- Burke H, Wilkinson TMA. 2021. Unravelling the mechanisms driving multimorbidity in COPD to develop holistic approaches to patient-centred care. *Eur Respir Rev* **30**. <http://dx.doi.org/10.1183/16000617.0041-2021>.
- Cavallès A, Brinchault-Rabin G, Dixmier A, Goupil F, Gut-Gobert C, Marchand-Adam S, Meurice J-C, Morel H, Person-Tacnet C, Leroyer C, et al. 2013. Comorbidities of COPD. *Eur Respir Rev* **22**: 454–475.
- Chen X, Zhou J, Zhang R, Wong AK, Park CY, Theesfeld CL, Troyanskaya OG. 2021. Tissue-specific enhancer functional networks for associating distal regulatory regions to disease. *Cell Syst* **12**: 353-362.e6.
- Cho MH, McDonald M-LN, Zhou X, Mattheisen M, Castaldi PJ, Hersh CP, Demeo DL, Sylvia JS, Ziniti J, Laird NM, et al. 2014. Risk loci for chronic obstructive pulmonary disease: a genome-wide association study and meta-analysis. *Lancet Respir Med* **2**: 214–225.
- Gim J, An J, Sung J, Silverman EK, Cho MH, Won S. 2020. A Between Ethnicities Comparison of Chronic Obstructive Pulmonary Disease Genetic Risk. *Front Genet* **11**: 329.

- GTEx Consortium. 2020. The GTEx Consortium atlas of genetic regulatory effects across human tissues. *Science* **369**: 1318–1330.
- Higbee D, Granell R, Walton E, Korologou-Linden R, Davey Smith G, Dodd J. 2021. Examining the possible causal relationship between lung function, COPD and Alzheimer's disease: a Mendelian randomisation study. *BMJ Open Respir Res* **8**. <http://dx.doi.org/10.1136/bmjresp-2020-000759>.
- Kim SH, Kim HS, Min HK, Lee SW. 2021a. Association between insulin resistance and lung function trajectory over 4 years in South Korea: community-based prospective cohort. *BMC Pulm Med* **21**: 110.
- Kim W, Prokopenko D, Sakornsakolpat P, Hobbs BD, Lutz SM, Hokanson JE, Wain LV, Melbourne CA, Shrine N, Tobin MD, et al. 2021b. Genome-Wide Gene-by-Smoking Interaction Study of Chronic Obstructive Pulmonary Disease. *Am J Epidemiol* **190**: 875–885.
- Kubysheva N, Soodaeva S, Novikov V, Eliseeva T, Li T, Klimanov I, Kuzmina E, Baez-Medina H, Solovyev V, Ovsyannikov DY, et al. 2018. Soluble HLA-I and HLA-II Molecules Are Potential Prognostic Markers of Progression of Systemic and Local Inflammation in Patients with COPD. *Dis Markers* **2018**: 3614341.
- Li C, Chen W, Lin F, Li W, Wang P, Liao G, Zhang L. 2022. Functional two-way crosstalk between brain and lung: The brain-lung axis. *Cell Mol Neurobiol*. <http://dx.doi.org/10.1007/s10571-022-01238-z>.
- Li Q, Zhao Q, Zhang J, Zhou L, Zhang W, Chua B, Chen Y, Xu L, Li P. 2019. The Protein Phosphatase 1 Complex Is a Direct Target of AKT that Links Insulin Signaling to Hepatic Glycogen Deposition. *Cell Rep* **28**: 3406-3422.e7.
- Machado FVC, Pitta F, Hernandez NA, Bertolini GL. 2018. Physiopathological relationship between chronic obstructive pulmonary disease and insulin resistance. *Endocrine* **61**: 17–22.
- Mak JCW, Chan-Yeung MMW, Ho SP, Chan KS, Choo K, Yee KS, Chau CH, Cheung AHK, Ip MSM, Members of Hong Kong Thoracic Society COPD Study Group. 2009. Elevated plasma TGF-beta1 levels in patients with chronic obstructive pulmonary disease. *Respir Med* **103**: 1083–1089.
- Mannino DM, Higuchi K, Yu T-C, Zhou H, Li Y, Tian H, Suh K. 2015. Economic burden of COPD in the presence of comorbidities. *Chest* **148**: 138–150.
- Niazi RK, Sun J, Have CT, Hollensted M, Linneberg A, Pedersen O, Nielsen JS, Rungby J, Grarup N, Hansen T, et al. 2019. Increased frequency of rare missense PPP1R3B variants among Danish patients with type 2 diabetes. *PLoS One* **14**: e0210114.
- Piazzolla G, Castrovilli A, Liotino V, Vulpi MR, Fanelli M, Mazzocca A, Candigliota M, Berardi E, Resta O, Sabbà C, et al. 2017. Metabolic syndrome and Chronic

Obstructive Pulmonary Disease (COPD): The interplay among smoking, insulin resistance and vitamin D. *PLoS One* **12**: e0186708.

- Platig J, Castaldi PJ, DeMeo D, Quackenbush J. 2016. Bipartite Community Structure of eQTLs. *PLoS Comput Biol* **12**: e1005033.
- Porcu E, Rüeger S, Lepik K, eQTLGen Consortium, BIOS Consortium, Santoni FA, Reymond A, Kutalik Z. 2019. Mendelian randomization integrating GWAS and eQTL data reveals genetic determinants of complex and clinical traits. *Nat Commun* **10**: 3300.
- Reimand J, Isserlin R, Voisin V, Kucera M, Tannus-Lopes C, Rostamianfar A, Wadi L, Meyer M, Wong J, Xu C, et al. 2019. Pathway enrichment analysis and visualization of omics data using g:Profiler, GSEA, Cytoscape and EnrichmentMap. *Nat Protoc*. <http://dx.doi.org/10.1038/s41596-018-0103-9>.
- Russ TC, Harris SE, David Batty G. RESEARCH LETTER: Pulmonary function and risk of Alzheimer dementia: two-sample Mendelian randomization study. https://discovery.ucl.ac.uk/id/eprint/10149131/2/Batty_FEV%20MR%20%28accepted%29.pdf (Accessed September 27, 2022).
- Sagun G, Gedik C, Ekiz E, Karagoz E, Takir M, Oguz A. 2015. The relation between insulin resistance and lung function: a cross sectional study. *BMC Pulm Med* **15**: 139.
- Sakornsakolpat P, Prokopenko D, Lamontagne M, Reeve NF, Guyatt AL, Jackson VE, Shrine N, Qiao D, Bartz TM, Kim DK, et al. 2019. Genetic landscape of chronic obstructive pulmonary disease identifies heterogeneous cell-type and phenotype associations. *Nat Genet* **51**: 494–505.
- Shrine N, Guyatt AL, Erzurumluoglu AM, Jackson VE, Hobbs BD, Melbourne CA, Batini C, Fawcett KA, Song K, Sakornsakolpat P, et al. 2019. New genetic signals for lung function highlight pathways and chronic obstructive pulmonary disease associations across multiple ancestries. *Nat Genet* **51**: 481–493.
- Steele-Perkins G, Plachez C, Butz KG, Yang G, Bachurski CJ, Kinsman SL, Litwack ED, Richards LJ, Gronostajski RM. 2005. The transcription factor gene *Nfib* is essential for both lung maturation and brain development. *Mol Cell Biol* **25**: 685–698.
- Stolz D. 2020. Chronic obstructive pulmonary disease risk: does genetics hold the answer? *Lancet Respir Med* **8**: 653–654.
- Szklarczyk D, Gable AL, Lyon D, Junge A, Wyder S, Huerta-Cepas J, Simonovic M, Doncheva NT, Morris JH, Bork P, et al. 2019. STRING v11: protein-protein association networks with increased coverage, supporting functional discovery in genome-wide experimental datasets. *Nucleic Acids Res* **47**: D607–D613.
- Verhamme FM, Bracke KR, Joos GF, Brusselle GG. 2015. Transforming growth factor- β superfamily in obstructive lung diseases. more suspects than TGF- β alone. *Am J Respir Cell Mol Biol* **52**: 653–662.

- Zaied R, Fadason T, O'Sullivan J. 2022. De novo identification of complex multimorbid conditions by integration of gene regulation and protein interaction networks with genome-wide association studies. *Research Square*. <http://dx.doi.org/10.21203/rs.3.rs-1313207/v1>.
- Zhou JJ, Cho MH, Castaldi PJ, Hersh CP, Silverman EK, Laird NM. 2013. Heritability of chronic obstructive pulmonary disease and related phenotypes in smokers. *Am J Respir Crit Care Med* **188**: 941–947.
- Zhu X, Duren Z, Wong WH. 2021. Modeling regulatory network topology improves genome-wide analyses of complex human traits. *Nat Commun* **12**: 2851.
- Zhu Z, Wang X, Li X, Lin Y, Shen S, Liu C-L, Hobbs BD, Hasegawa K, Liang L, International COPD Genetics Consortium, et al. 2019. Genetic overlap of chronic obstructive pulmonary disease and cardiovascular disease-related traits: a large-scale genome-wide cross-trait analysis. *Respir Res* **20**: 64.

November 22, 2022

RE: Life Science Alliance Manuscript #LSA-2022-01609-TR

Dr. Justin M. O'Sullivan
The University of Auckland
The Liggins Institute
University of Auckland
Private Bag 92019
Auckland 1142
New Zealand

Dear Dr. O'Sullivan,

Thank you for submitting your revised manuscript entitled "De novo discovery of traits co-occurring with chronic obstructive pulmonary disease". We would be happy to publish your paper in Life Science Alliance pending final revisions necessary to meet our formatting guidelines.

- please address the final Reviewer #2's points
- please upload both your main and supplementary figure legends as single files; please add a separate figure legend section to your manuscript text, including the main figure legends, supplementary figure legends, and the table legends

A. FINAL FILES:

B. MANUSCRIPT ORGANIZATION AND FORMATTING:

Sincerely,

Reviewer #1 (Comments to the Authors (Required)):

The authors have sufficiently addressed the concerns raised in the review.

Reviewer #2 (Comments to the Authors (Required)):

1.-A short summary of the paper, including description of the advance offered to the field.

Chronic obstructive pulmonary disease (COPD) is a complex disease, with environmental but also genetic factors as well as known to co-occur with other conditions. Golovina et al present here a very interesting method to study and better characterise the diversity of this disease: they started by expanding common variants from GWAS to lung specific eQTLs and integrating this information with a lung-specific gene regulatory network and protein protein interaction network. Together with common variation from all traits contained in the GWAS catalog, they can study the biological mechanisms behind co-occurrence, reaching a better understanding of COPD, needed for improvements in the clinics (better stratification and better treatments).

2.-For each main point of the paper, please indicate if the data are strongly supportive. If not, explicitly state the additional experiments essential to support the claims made and the timeframe that these would require.

My main concerns raised during the first round have been address: the network analysis is amended, and the discussion changed accordingly (points A, C and D of my original comments), the connexion with Bin1 is now excluded (point B) and the minor comments including typos or suggestions in the tables are also updated. I appreciate the additional lines in the discussion (point 4 of the rebuttal letter), enhancing the knowledge supporting the connexion with insulin resistance and immune pathways.

3.-Lastly, indicate any additional issues you feel should be addressed (text changes, data presentation, statistics etc.).

In page 5 of the manuscript, we can find the following sentence "as being significantly ($p < 0.05$) enriched within the 107 COP associated genes (Supplemental Table S4, Supplemental Fig. S4)." According to the supplementary table, this value should be corrected for multiple testing. I suggest including here that information in the same way it is stated some lines below: "Pathway analysis of these 107 COPD-associated genes identified biological pathways that were enriched ($FDR < 0.05$)".

I have also a small comment concerning the networks display (supplementary figure 5). The meaning of the network edges is set as "evidence", I strongly suggest changing to "confidence" instead, there is no particular reason to show the evidence (it is not part of the discussion) and it creates unnecessary thick lines (specially detrimental for supplementary figure 5). More important, as it is now it is very difficult to identify the 3 interacting genes from figure 4 and how they interact with the "NUPR1-MSL1-SGF29-MOCS2" axes. I suggest underlining this connexion (maybe different colouring for the edges and the nodes?) as it is a main point of the discussion and important information concerning figure 4.

Reviewer 2:

1. In page 5 of the manuscript, we can find the following sentence "as being significantly ($p < 0.05$) enriched within the 107 COP associated genes (Supplemental Table S4, Supplemental Fig. S4)." According to the supplementary table, this value should be corrected for multiple testing. I suggest including here that information in the same way it is stated some lines below: "Pathway analysis of these 107 COPD-associated genes identified biological pathways that were enriched ($FDR < 0.05$)".

➤ We have modified the text as follows:

Functional gene ontology enrichment analysis identified metabolic, behavioural, regulatory and protein modification processes (e.g. "phosphorus metabolic process", "behavioral response to nicotine", "regulation of postsynaptic membrane potential", "protein acetylation" and "protein acylation") as being significantly enriched ($FDR < 0.05$) enriched within the 107 COPD-associated genes (Supplemental Table S4, Supplemental Fig. S4).

2. I have also a small comment concerning the networks display (supplementary figure 5). The meaning of the network edges is set as "evidence", I strongly suggest changing to "confidence" instead, there is no particular reason to show the evidence (it is not part of the discussion) and it creates unnecessary thick lines (specially detrimental for supplementary figure 5). More important, as it is now it is very difficult to identify the 3 interacting genes from figure 4 and how they interact with the "NUPR1-MSL1-SGF29-MOCS2" axes. I suggest underlining this connexion (maybe different colouring for the edges and the nodes?) as it is a main point of the discussion and important information concerning figure 4.

➤ We have modified Supplemental Figure 5 to include confidence lines, in response to the referee's comment.

December 8, 2022

RE: Life Science Alliance Manuscript #LSA-2022-01609-TRR

Dr. Justin M. O'Sullivan
The University of Auckland
The Liggins Institute
University of Auckland
Private Bag 92019
Auckland 1142
New Zealand

Dear Dr. O'Sullivan,

Thank you for submitting your Research Article entitled "De novo discovery of traits co-occurring with chronic obstructive pulmonary disease". It is a pleasure to let you know that your manuscript is now accepted for publication in Life Science Alliance. Congratulations on this interesting work.

DISTRIBUTION OF MATERIALS:

Again, congratulations on a very nice paper. I hope you found the review process to be constructive and are pleased with how the manuscript was handled editorially. We look forward to future exciting submissions from your lab.

Sincerely,
